# Systematic review and meta-analysis for a Global Patient co-Owned Cloud (GPOC)

Niklas Lidströmer [1,2] ✉, Joe Davids[3], Mohamed ElSharkawy [3], Hutan Ashrafian [3] & Eric Herlenius [1,2]

Cloud-based personal health records increase globally. The GPOC series introduces the concept of a Global Patient co-Owned Cloud (GPOC) of personal health records. Here, we present the GPOC series' Prospective Register of Systematic Reviews (PROSPERO) registered and Preferred Reporting Items Systematic and Meta-Analyses (PRISMA)-guided systematic review and meta-analysis. It examines cloud-based personal health records and factors such as data security, efficiency, privacy and cost-based measures. It is a meta-analysis of twelve relevant axes encompassing performance, cryptography and parameters based on efficiency (runtimes, key generation times), security (access policies, encryption, decryption) and cost (gas). This aims to generate a basis for further research, a GPOC sandbox model, and a possible construction of a global platform. This area lacks standard and shows marked heterogeneity. A consensus within this field would be beneficial to the development of a GPOC. A GPOC could spark the development and global dissemination of artificial intelligence in healthcare.

The concept of a Global Patient co-Owned Cloud (GPOC) embodies a global and blockchain protected, worldwide distributed and patient co-owned platform of personal health records (PHR, ISO/TR 14292:2012). Until now, this concept of a co-ownership model on a global scale has not been presented.

Here, the GPOC series commences with a systematic review and meta-analysis of a dozen pivotal facets of a GPOC. It aims to cover the dozen facets most relevant to the technical construction of a GPOC model.

The GPOC series consists of four other self-contained publications[1–4]. The GPOC concept's necessity is explored in the GPOC Survey, revealing a global consensus[1]. This received answers from all key opinion leaders of 193 + 3 United Nations' member states and the 18 largest international health care organisations[1]. Thus, the technical and mathematical foundations were shaped, resulting in a GPOC sandbox environment[2].

Cloud-based PHRs have become increasingly vital in healthcare, enhancing patient management. The quality of patient care hinges on maintaining data integrity, privacy, security, and efficient data retrieval for clinicians and healthcare providers[5,6]. Centralised PHRs have faced criticism for security vulnerabilities and clinician burnout[7]. For instance, the WannaCry ransomware attack, which began in 2017 and continues to pose a threat, targets less secure central systems. It affected over 150 countries and over 40% of the world's national health care systems[8]. In the security evolution new cloud-based models, including blockchain-based systems, have been researched worldwide[7]. These offer enhanced privacy, security, and access control. Some even allow for the deletion of patient information when necessary, addressing privacy concerns[5,6].

Another issue arises with travellers in a globalised world, as their healthcare records may not be accessible in host nations. This underscores the need for a secure cloud-based global PHR platform that can support both patient care during travel and migration.

Ensuring the security of these cloud-based PHRs involves advanced cryptographic techniques, necessitating continuous research and testing. However, emerging technologies also pose

[1]Department of Women's and Children's Health, Karolinska Institutet, CMM, L8:01, 17176 Stockholm, Sweden. [2]Astrid Lindgren Children's Hospital, Karolinska University Hospital, Stockholm, Sweden. [3]Institute of Global Health Innovation and the Hamlyn Centre for Robotic Surgery, Imperial College London, London, UK. ✉ e-mail: niklas.lidstromer@ki.se

regulatory and ethical challenges, especially regarding data ownership and responsibility[4].

Here, the systematic review and meta-analysis explore the impact of these technologies on the concept of co-ownership across borders. Hereby enabling a foundation to assess PHR management and design for a global patient co-owned cloud.

## Results

### Overview

The PRISMA flow diagram in Fig. 1 summarises the screening process. Search results retrieved 16,045 references with 6683 duplicates removed and 9362 references screened. Thirty-four were selected for final inclusion in the review and 12 were included in the meta-analysis. Figure 2 depicts the twelve GPOC core facets included in the systematic review and meta-analysis. Figure 3 shows the geographical distribution of the institutions included in the GPOC systematic review and meta-analysis. As an illustration of our analytical approach, Fig. 4 showcases a forest plot derived from the meta-analysis, while all forest plots are available in Supplementary File 2 (S2).

### Efficiency-based parameters

Runtimes defines the amount of time it takes for a programme or piece of code to run (ms). In 117 sub studies on runtimes, a pooled effect size estimate of 12874 ms (CI: 12867–12881, $I^2$ 100%; $p = 0.0005$). A log transformed meta-analysis of the 117 sub studies on runtimes also showed an effect size estimate of 1.98 ms (CI: 1.97–1.98, $I^2$ 100%; $p = 0.0005$).

Key generation times was defined as the time required for the process of generating cryptographic keys (ms). In 46 sub studies on key generation time, a pooled effect size estimate of 143 ms (CI: 121–165, $I^2$ 98%; $p = 0.0005$). A log transformed meta-analysis of the 46 sub studies on key generation time also showed an effect size estimate of 4.5 ms (CI: 4.52–4.47, $I^2$ 99.9%; $p = 0.0005$). Figure 4 illustrates the forest plot for the key generation time meta-analysis.

### Other time-based activities

In 26 sub studies on time analysis such as key management and increased keyword query search time for PHR server transfer, a pooled effect size estimate of 3951 ms (CI: 3949–3955 $I^2$ 100%; $p = 0.0005$). A log transformed meta-analysis of the 26 sub studies on usage policy also showed an effect size estimate of 2.56 ms (CI: 2.55–2.56, $I^2$ 100%; $p = 0.0005$).

### Security-based parameters

Access policies define the protection of cloud data access and devices. These are set up to block access to all unauthorised uploads. In 34 sub studies on usage policy, a pooled effect size estimate of 30076 security-based policy of granularity of data access and response (CI: 30073–30079, $I^2$ 100%; $p = 0.0005$) was identified. A log transformed meta-analysis of the 34 sub studies on usage policy also showed an effect size estimate of 3.98 policies (CI: 3.97–3.98, $I^2$ 100%; $p = 0.0005$).

Encryption ensures the conversion of information secretly to hide its original contents and was defined as the total encrypted data (bytes) divided by the encryption time (ms). In 86 sub studies on encryption, a pooled effect size estimate of 80.76 ms (CI: 80.7–80.7, $I^2$ 100%; $p = 0.0005$). A log transformed meta-analysis of the 86 sub studies on encryption also showed an effect size estimate of 1.86 ms (CI: 1.86–1.86, $I^2$ 100%; $p = 0.0005$).

In 20 sub studies on ratio of means of encryption, a pooled effect size estimate of 0.16 ms (CI: 0.11–0.21, $I^2$ 100%; $p = 0.0005$). A log transformed meta-analysis of the 20 sub studies on ratio of means of encryption also exhibited an effect size estimate of 0.162 ms (CI: 0.110–0.214, $I^2$ 100%; $p = 0.0005$).

Decryption reverses the coded information to its original content and was defined as the total decrypted data (bytes) divided by the decryption time (ms). In 73 sub studies on decryption, a pooled effect size estimate of 59.50 ms (CI: 59.50–59.51, $I^2$ 100%; $p = 0.0005$). A log transformed meta-analysis of the 73 sub studies on decryption also showed an effect size estimate of 1.70 ms (CI: 1.70–1.70, $I^2$ 100%; $p = 0.0005$).

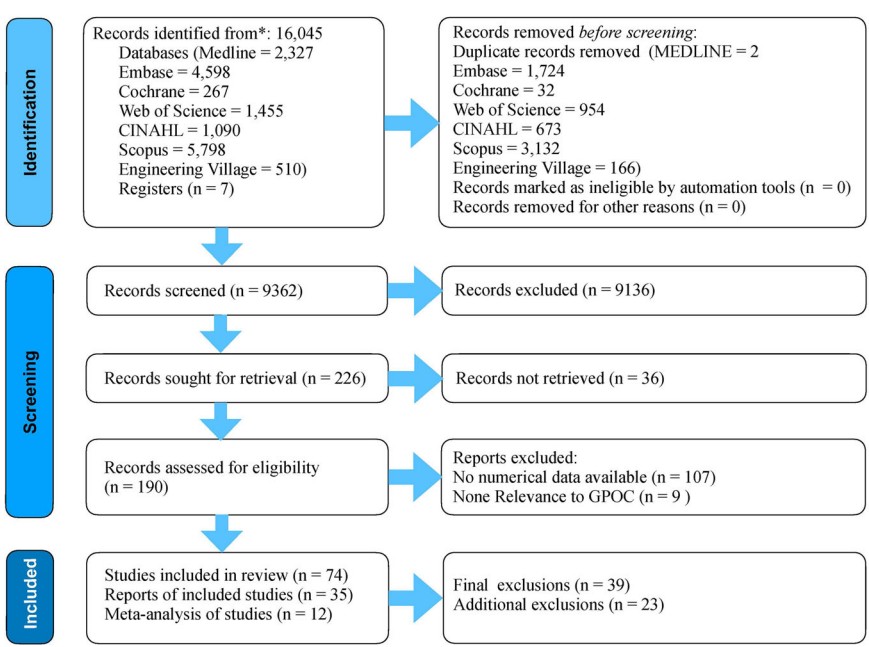

**Fig. 1 | Search strategy and article selection process: PRISMA flowchart.** PRISMA Flow chart illustrating our search strategy and article screening and selection. For the PRISMA 2020 checklist see Supplementary File 3 (S3). The chart was created with KeyNote 11. Source Data files are available in the article repository on Figshare, https://doi.org/10.6084/m9.figshare.c.7066553.

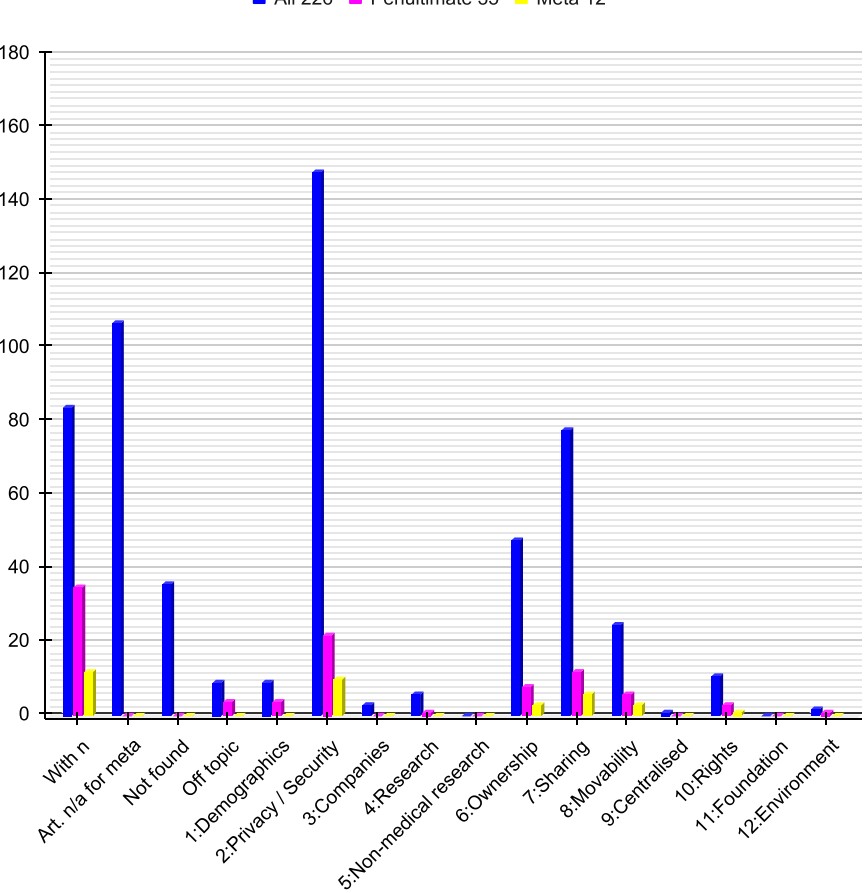

**Fig. 2 | Overview of core subjects in retrieved articles.** Overview of the twelve core subjects included in the 226 articles sought for retrieval (blue), the penultimate 35 articles (pink), and the 12 meta-analysed articles (yellow). Eighty-four articles (37%) contained in numbers, 107 articles contained no numerals (47%), 36 articles were not retrieved (16%), and 9 articles had no relevance to GPOC (4%). Of all 226 articles, the top-three subjects were privacy/security covered by 148 articles (65%), sharing by 78 (35%), and ownership by 48 articles (21%). Of the penultimate 35 articles, the top-three subjects were privacy covered by 22 articles (63%), sharing by 12 articles (34%), and movability by 6 articles (17%). Of the meta-analysed 12 articles the top-three subjects were privacy covered by 10 articles (83%), sharing by 6 articles (50%), and movability by 3 articles (25%). For the GPOC Word Cloud of the 100 commonest words, based on 38,000 words selected equally and representatively from all 190 eligible articles (out of 226 articles sought for retrieval), see Supplementary File 4 (S4). Source Data files are available in the article repository on Figshare, https://doi.org/10.6084/m9.figshare.c.7066553.

## Cost-based parameters

Data transfer cost (gas cost) was defined as gas, which is the price per unit of computation that is performed on the Ethereum network. In 8 sub studies on gas analysis, a pooled effect size estimate of 70193 Ethereum (CI: 70113–70272, $I^2$ 100%; $p = 0.0005$). A log transformed meta-analysis of the 8 sub studies on gas analysis also showed an effect size estimate of 1.71 Ethereum (CI: 1.63–1.79, $I^2$ 99.9%; $p = 0.0005$).

## Risk of Bias (ROB)

Figure 5 illustrates risk of bias of the 12 meta-analysed studies across seven bias domains, with 31% moderate and 69% of low risk. The studies presented moderate risks of bias: 8% due to confounding, 75% due to selection of participants, 25% in classification of interventions, 42% due to deviations from intended interventions, 25% due to missing data, 17% in measurement of outcomes, and 25% in selection of the reported result.

## Discussion

Cloud-based PHRs gain momentum worldwide. This motivates research into their security, managing, efficiency, and costs. This field has never been meta-analysed before. The findings provide the foundation for the eventual construction of a GPOC. A global PHR platform could power machine learning and spark AI within healthcare everywhere. Though, PHR datasets remain fragmented. There are also many ethical, policy and regulatory challenges. For instance, security and security have implications of HIPAA and GDPR. These are analysed on a global scale in another part of the GPOC-series[4].

In addition to centralised PHRs, there are alternative architectures. These include fog-based, peer-to-peer and hierarchical methods. The former leverage edge computing resources, providing proximity benefits and enhancing data privacy. The two latter models distribute control and ownership among users. With these a GPOC could offer greater autonomy.

The integration of AI into GPOC would provide incorporated multilingual support and patient decision guidance. Bridging language and some education barriers. Notably, AI integration might interpret and explain complex medical texts to patients, interact and provide advice. Hence, a medical GPOC integrated generative AI. Likewise, patients with impaired hearing or vision would also be helped with integrated AI tools. Here, natural language processing (NLP) would provide in real-time assistance and decision support to co-owners.

Currently, these AI tools are often made by companies. They have trained algorithms on data. However, patients' consent is pivotal[4,9]. Therefore, integrated omics data for precision medicine provides both possibilities and considerations[4,10].

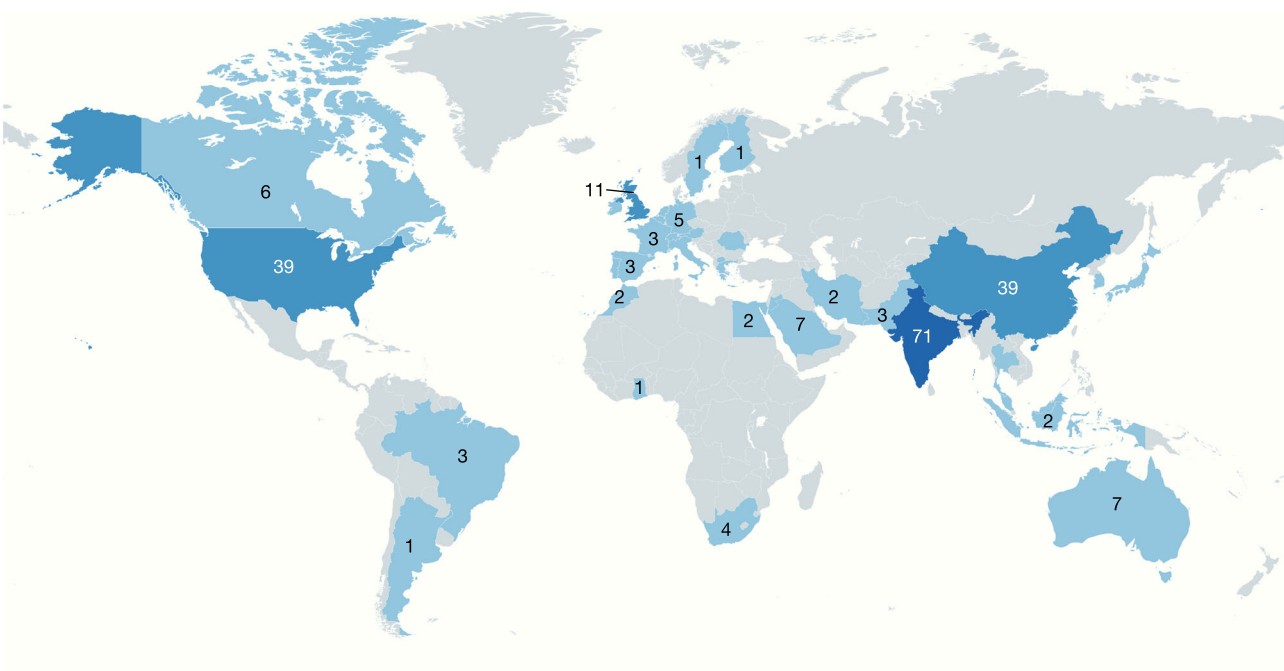

**Fig. 3 | Global distribution of institutions and gender representation in GPOC study authors.** Illustration of the global breakdown and distribution of the institutions in the 47 countries of 834 co-authors of the 226 articles included in the GPOC systematic review and meta-analysis. 42% of the 1st authors were women. The map was created using MapChart.net and adapted with KeyNote 11. Source Data files are available in the article repository on Figshare, https://doi.org/10.6084/m9.figshare.c.7066553.

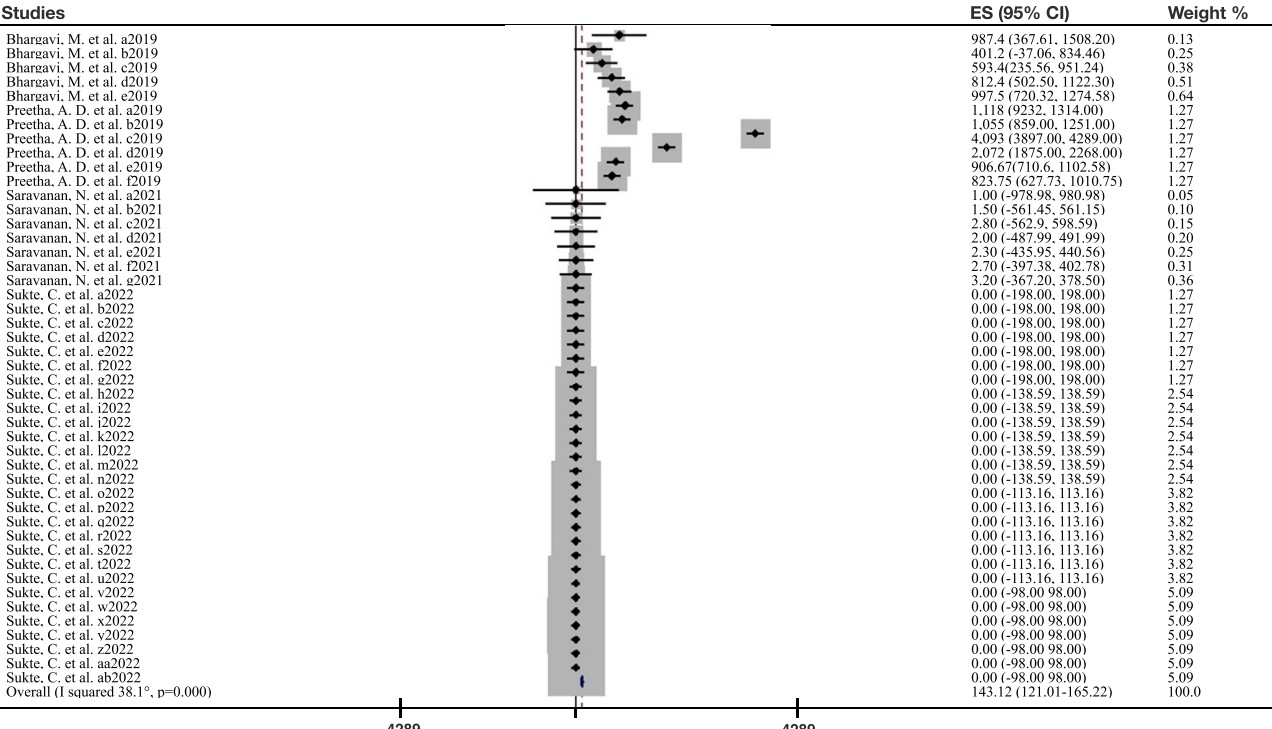

**Fig. 4 | Forest plot for key generation time meta-analysis.** The forest plot displays the results of the meta-analysis for key generation time, with a heterogeneity chi-squared of 2430 (degrees of freedom 45), $p = 0.0005$, and I-squared (variation in effect size attributable to heterogeneity) of 98.1%. ES (effect size) with 95% confidence interval (CI) is shown. The diamonds represent the pooled effect size estimates, with error bars indicating the 95% confidence intervals. Please note that the measure of centre for the error bars corresponds to the mean of each estimate. Statistical tests used were two-sided. The forest was created with Stata 17 and adaped with KeyNote 11. For all 18 forest plots of the meta-analysis, refer to Supplementary File 2 (S2). Source Data files are available in the article repository on Figshare, https://doi.org/10.6084/m9.figshare.c.7066553.

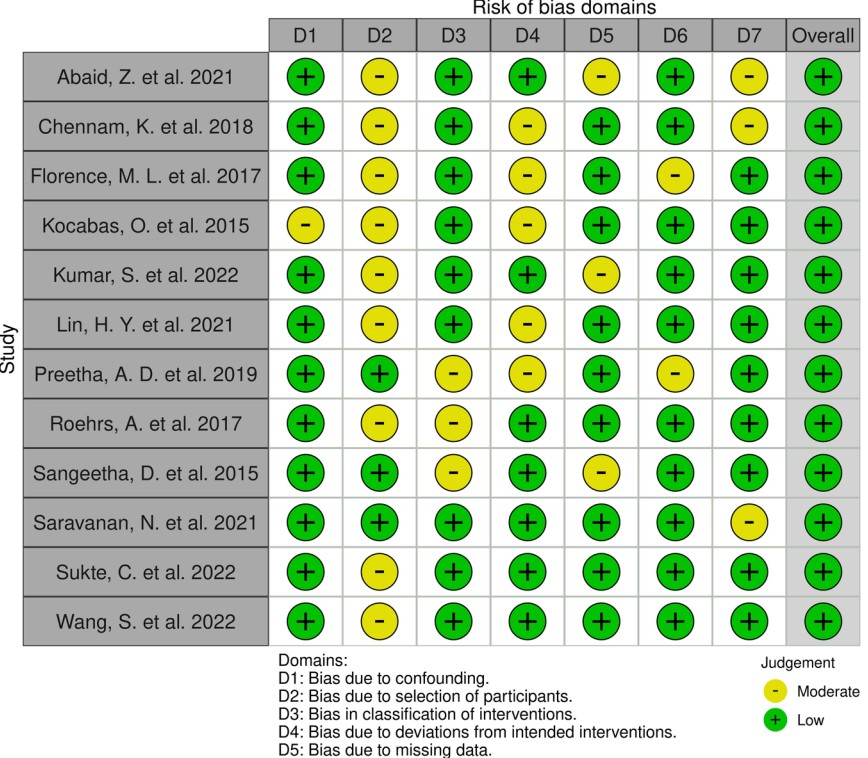

**Fig. 5 | Assessment of risk of bias in meta-analysed studies.** Pictorial representation of the results from the risk of bias analysis indicating low to moderate risk of bias presented in the studies[14–20,23–26,39]. The figure was generated with the online RobVis tool. Source Data files are available in the article repository on Figshare, https://doi.org/10.6084/m9.figshare.c.7066553.

The results' section showed security-based parameters such as encryption and decryption time in milliseconds (ms). However, these metrics are inherently influenced by the basic infrastructure. This includes CPU performance, available memory, network bandwidth, and other hardware resources. The absolute timing values may not give the full picture of security issues. For instance, a more powerful CPU or increased network bandwidth, might reduce time and enhance security. Therefore, the results must be put into context and industry standards considered.

Here, efficiency-based parameters that could impact the retrieval of data from PHR were meta-analysed. Significantly fast run-times for PHRs were seen in 117 sub studies. These studies demonstrate that information access speed may impact clinical decision-making. Communication within healthcare would benefit from accelerated models.

One study on efficiency and performance of cloud-based PHRs analysed chunking, bundling, deduplication, delta-encoding, and data compression[11]. All these contribute to cloud characteristics. Performance indicators included control data overhead quantification of an average packet transmission rate of 93% compared to cloud storage services[11]. This illustrates one way of comparing several factors in parallel, to find a suitable PHR solution.

Others have shown similar results. That is, with a time efficiency comparison of values for different attributes, with different files ($n = 10$–$50$, and key generation times 401–998 ms)[12]. Computation times ($|S| = 100$) have been presented for six models (WTCM, WTCF, SADS, VAKF, LSTM, MLPPT-MHS), with key generation times ranging 824–4093 ms[13,14]. Thus, a great variation is seen. A computation of delegation in key verification by comparing two models (proposed ECP-ABE vs existing CP-ABE), gave key generation times varying from 1 vs 1.2 ms to 3.2 vs 4.7 ms[15,16]. A comparison of six models (Blowfish, RSA, ASE, El-Gamal, ECC, Modified El-Gamal, Modified ECC) showed key generation times from 1–14 ms[17]. Others argue that centralised cloud providers to organisations affect the ease of movement of various PHR datasets[18]. The ease of moment is important for a GPOC.

The above-mentioned technical variables may affect how effectively PHRs are shared. Possibly patients may be open to sharing for better care and for research, even in the face of privacy concerns[19].

The security efficiency of PHRs was presented with encryption types and how cryptographic keys are generated to safeguard from unauthorised access. PHRs were significantly more efficient than other methods of record keeping. However, this efficiency can pose risk to PHR integrity and lead to ransomware attacks or distributed denial of service (DDoS) attacks. Similarly, other time-based analysis including file transfer times had a significantly better efficiency-based measure recorded for PHRs[20].

PHR security with access policies, optimised speed and efficiency of shared data is therefore pivotal. To measure these objectively is hard. Many users do not have technical knowledge and may not be aware of security risks leading to unauthorised access. Here, we identified a heterogenous pooled effect size estimate of 30,076 security policies, that impact granularity of data access and response. An effect size estimate of 3.98 policies ($p = 0.0005$) in log transformed meta-analysis was also identified here. This data is novel.

However, discussions exist regarding innovations, such as patient-controlled health access brokerage services with necessity for implementing security logs and unique methods of intrusion detection[20–22].

Encryption is a backbone of security. Security concerns were discussed by a majority of included articles. With 148 articles (65%) it was the commonest elaborated of all facets. The encryption type, is crucial for safe transfer, sharing and compressing PHRs. In 86 substudies on encryption, a pooled effect size estimate of encryption speed of 81 ms was seen. The response ratio was examined on 20 substudies, looking at mean encryption times, which demonstrated an effect size estimate of 0.16 ms. The literature review of the meta-analysed articles presented

several studies on encryption (800–1200 ms[20], 8654-10025 ms[23], 29-98 ms[24], 80-5040 ms[25], 9919–280 ms[12], 8–12 ms[16], and one team compared six schemes (Blowfish, RSA, ASE, El-Gamal, ECC, Modified El-Gamal, and Modified ECC) with ranges 0.00006–0.03 ms[17], and where the meta-analysis gave a pooled effect size estimate of 81 ms, an effect size estimate of 1.86 ms and $p = 0.0005$).

Decryption time, necessary for retrieving information by a patient or clinician, had a pooled effect size estimate of 59.5 ms. This is a PHR benchmark, which future studies could improve. Several studies presented decryption, e.g., 4236–7546 ms[20,23], 16–74 ms[24], 30–2290 ms[25], 4–12 ms[26], 90–71,167 ms[12]. Moreover, one team compared six schemes (Blowfish, RSA, ASE, El-Gamal, ECC, Modified El-Gamal, and Modified ECC) with ranges 0.000086–0.00054 ms[17], The meta-analysis gave a pooled effect size estimate of 59.5 ms, an effect size estimate of 1.7 ms and $p = 0.0005$) performances for proposed algorithms and security solutions[27–31].

There are several encryption types. For instance, symmetric encryption uses a single key for both encryption and decryption and is known for its speed and efficiency. Asymmetric encryption, also called public-key cryptography, uses a pair of keys (public and private) for secure communication. Homomorphic encryption allows users to perform computations on encrypted data without the need to decrypt it, preserving privacy. Hence, a fully homomorphic encryption (FHE) allows users to analyse on encrypted datasets without seeing the underlying data[32]. For GPOC we have explored this type further[2]. End-to-End Encryption is often used in communication applications, this ensures that only the sender and recipient can access the content, making it highly secure. Blockchain-embedded PHRs utilises blockchain encryption, ensuring data immutability and security through distributed, tamper-proof and interoperable ledgers. Here patients can regulate PHR access. These encryption protocols enhance security, traceability and privacy of PHRs, explored further in the technical part of the GPOC series[2].

While encryption and decryption are crucial aspects of security, a more holistic analysis must contain other fundamental pillars. In addition, confidentiality means examining access controls, user authentication and data masking techniques that protect PHRs from unauthorised access. Ensuring integrity, involves digital signatures, checksums and audit trails hindering PHR tampering or alteration. In healthcare continuous availability of PHRs may be lifesaving. This includes redundancy in data storage, disaster recovery plans and load balancing strategies. Several articles discussed confidentiality (19), integrity (14) and availability (12). However, these were not measured enough with numeric values for a meta-analysis.

Previous proof-of-work blockchain technologies allowed an organisation to calculate exact costs of performing software and mathematical operations needed for digital tokenisation and activities. This is expressed as gas on Ethereum, which is a decentralised blockchain with smart contract functionality. This has advantages, since those operating on the Ethereum virtual machine use a measurable gas cost for executing programmes supporting the functioning of the PHR, using smart executable contracts. Based on inborn technical limitations of the design standard for smart contracts, these could be tailored to one specific action without affecting other necessary components of the PHR. This makes useability costs measurable and auditable. The gas meta-analysis demonstrated a pooled effect size estimate of 70193 ms with a log transformed effect size estimate of 1.7 ms. An ideal PHR should allow accurate estimation of costs for information transfer, data mining and interdisciplinary access for decision support to compensate users in a co-ownership model.

While Blockchain technology is a significant trend, the field of PHRs moves rapidly. Emerging technologies such as Federated Learning, Fog Computing and the Internet of Things are poised to shape future PHRs. Blockchain's decentralised and immutable ledger capabilities continue to spread among PHRs and improve security,

integrity and interoperability. Federated Learning allows collaborative training in distributed datasets while upholding privacy. It will potentially revolutionise how PHRs will be used for research and enable personalised precision healthcare without centralisation. Fog computing extends the edge computing capabilities. It enables real-time data processing at the edge of the network. Hence, it enhances the responsiveness of PHRs. This would advance the field of critical applications such as remote monitoring. With Internet of Things (IoT) data from wearables and devices could be integrated with PHRs. This data convergence takes real-time monitoring and personalised care to a new level. The above technologies represent pivotal health IT trends, with important applications and synergies with cloud-based PHRs.

One study applied Blockchain technologies in patient-centric models for PHR data management allow for smarter interconnectivity between healthcare and the Internet of Things (IoT)[33]. The aim is to streamline the provision of higher quality privacy powered healthcare services using zero-knowledge proofs. The intended consequence is a fusion of a zero-knowledge proof for encryption whilst ensuring patient consent is acquired for data insight discovery to maintain privacy and anonymity. One patient-centred PHR model with an information access control scheme used Lagrange interpolation polynomials for secure multi-user permissible information access[21]. Many teams discuss the application of machine learning analysis of cloud-based datasets and IoT[34]. Also the driving development role of companies' AI tools for large datasets[35].

In all future healthcare, machine learning will play a central role. Data-driven decision-making in healthcare may be integrated into a GPOC and needs several methods to preserve patient privacy. Anonymisation, such as de-identification and pseudonymization, play a pivotal role in protecting patient identities while enabling PHR for research. These methods help mitigate privacy concerns associated with data sharing and analysis. Obfuscation involves the transformation of sensitive data to protect the confidentiality, while still allowing meaningful analysis. It is an effective means to strike a balance between data utility and privacy protection.

The significant global trend of interoperability means that different PHR systems and software applications could seamlessly exchange patient data across platforms and organisations. It is crucial to improve patient care and streamline the administration, and boost both research and AI development. To make this easier, there are technical standards and policies have been developed. An important example is HL7 FHIR (Fast Healthcare Interoperability Resources). It is an open standard for healthcare data exchange that focuses on simplicity, flexibility, and scalability. It uses RESTful web services and resources to enable the exchange of structured clinical and administrative data. FHIR resources are designed to represent specific healthcare concepts. These use widely accepted healthcare terminologies, facilitating sharing. FHIR also incorporates modern web technologies, such as JSON and XML, to help developers.

In addition, other technical standards and policies include: HL7 v2.x, CDA (Clinical Document Architecture), DICOM (Digital Imaging and Communications in Medicine), IHE (Integrating the Healthcare Enterprise), HIPAA (Health Insurance Portability and Accountability Act) and EHR Certification Programmes.

Finally, for three time-based security aspects there are neither previous meta-analyses, nor standards: (1) Runtimes. In a GPOC these could reduce the effects of data retrieval lag. In 117 substudies on runtimes, a pooled effect size estimate of 12874 ms (CI 12867–12881, $I^2$ 100%; $p = 0.0005$). A log transformation gave 1.98 ms (CI: 1.97–1.98, $I^2$ 100%; $p = 0.0005$). (2) Key generation times. In 46 substudies on key generation time, a pooled effect size estimate of 143 ms (CI: 121–165, $I^2$ 98.1%; $p = 0.0005$). A log transformation gave 4.49 ms (CI: 4.52–4.47, $I^2$ 99.9%; $p = 0.0005$). (3) Server transfer times. In 26 substudies on time analysis, such as key management and increased keyword query search time for PHR server transfer, a pooled effect size estimate of 3952 ms

(CI: 3949–3955, I² 100%; *p* = 0.0005). A log transformation gave 2.56 ms (CI: 2.55–2.56, I² 100%; *p* = 0.0005). Thus, there are no previous meta-analyses or standards for three time-based security aspects, highlighting the need for further research in these areas.

In summary, there are several future key challenges:

1. Global Healthcare Data Platform: Future efforts should focus on designing a comprehensive global PHR platform to combat health crises and promote global health. This platform would enable international healthcare and research communication and interaction. During COVID-19, researchers tried to design a global pandemic monitoring platform[36]. Others conclude that the present centralised systems cannot adapt to the vast volumes of globalised PHRs[6]. An optimal and complete use of PHRs could become prophylactic and have a major impact on global health[37]. Another team concludes that COVID-19 a global PHRs platform, would play a pivotal role in combatting the pandemic[38].

2. AI Integration and Security: Siloed use of AI on health data, security concerns and no pipeline for future AI improvement[12]. Future work should explore integrated AI-empowered cloud-based PHR systems. Patients sharing their PHR contents and usage of AI on their data is a game changer[39]. An AI-empowered cloud-based PHR system, which could possibly decrease healthcare errors, costs, and improve quality and effectiveness has been suggested[40]. Although PHRs facilitate healthcare, these are often outsourced to third party cloud service providers, bringing severe security issues, and increasing the risk of malicious usage and leakages[41].

3. User Experience and User Interface (UX/UI): Current PHRs are non-interactive and lack ergonomic user interface. Studies have shown they are so badly designed that it causes health worker burnouts[7]. The design must be user-friendly with elderly tools integrated. It should be possible to integrate IoT and AI tools. Importantly, cloud-based PHRs may become simplified health sharing platforms[42]. For instance, sharing could be to friends, family or professionals. A team presented the Bluefish algorithm to improve the security, flexibility, and transmission to third-party cloud providers[43,44]. At present cloud security solutions cannot handle all sophisticated threats[39]. There are proposed re-encryption solutions in response to white-box attacks. This to maintain efficiency even if there are multiple recipients. Easy accessibility and straight provider access as key vulnerabilities have been identified[45].

4. High PHR Software Costs: PHRs are too expensive for many health economies globally. There are economic and access advantages with cloud based PHR platforms[46]. Even though cloud storage can cut costs and improve health data sharing, the security issues are still substantial[21].

5. Effective Use of Health Data: Presently PHRs are hindering effective use of health data. This impacts AI progress in medicine. Multi-source PHRs with socioeconomic and genetic data would advance precision healthcare.

6. Global Adoption of PHRs: Globally relevant ethnic and social perspectives of the patient journey and PHR adoption have been studied[47]. As a continuation to these, the needs of the disabled persons from ethical, social, and judicial perspectives, have been elaborated[48]. Another team also showed how multi-source PHRs with both socioeconomic and genetic data will have a pivotal role in the realisation of true global and individual-centred precision healthcare of the near future[49].

7. Interaction and Communication: Lack of interaction and communication leads to one fifth of PHRs having serious errors. Current PHRs are costing time, money, and lives[50]. Patients' self-management of PHRs has been suggested, along with control and full ownership[22]. It has been suggested this decreases the amount of PHR errors with less nosocomial and adverse effects. Health expenses are rising with an older global population, and an intelligent cloud-based electronic health record (ICEHR) has been suggested to diminish medical mistakes[40]. Another concept is the individual-focused, long term and 'error-free' PHR[47]. Another project involves a smartphone application with a self-administrative medical solution aiming at increasing PHR correctness[51].

8. Global Patient co-Owned Cloud (GPOC): A GPOC would mean a global and AI empowered platform which would be a solution to the mentioned challenges. It has also been discussed how a GPOC could be self-sufficient, and hence facilitate global dissemination of PHRs and AI for global health[2,4]. Moreover, a GPOC in the form of a foundation has been discussed[1,3]. The ethics' article in the GPOC series concludes, among other, the necessary trisection of ownership between patients, clinicians and clinics[4].

This study is the first in the field. There is no standard yet. Hence, a clear heterogeneity. It was controlled for using a random effect model. The results were significant within core aspects of PHR security, efficiency and cost.

Future research may involve the collaboration of stakeholders to develop a consensus-driven approach to standardize PHR data. This would support effective and secure access for clinicians and organisations. It could also enable a standardised approach for AI integration into a future GPOC.

### Final remarks
In conclusion, the meta-analysis of twelve axes for a future GPOC currently demonstrates marked heterogeneity. This is a consequence of a new field without standards. Although we have meta-analysed the cryptographic, cost, performance and speed of the basic techniques that are currently available. This would facilitate the construction of a GPOC. We have highlighted several limitations. A consensus may come within the field of privacy and security for cloud based Blockchain PHRs. The eventual GPOC may benefit global health.

## Methods
### Search strategy
The PRISMA-guided multi-platform database review was registered on PROSPERO (CRD42022342597). Supported by librarians of Karolinska Institutet and Imperial College London. Thematic keyword searches on Ovid Medline, PubMed, Cochrane Library, EMBASE, Web of Science core collection, CINAHL, SCOPUS and Engineering village (Inspec and Knovel). The overarching themes were global cloud-based, decentralised, patient co-ownership, personal/electronic health record systems. Keywords included data co-ownership, patient rights, artificial intelligence, ethics, data infrastructure, economics, regulatory, patient outcomes, and auditing. The period ranged 1946–2022. For complete search strategy see Supplementary File 1 (S1).

### Screening
Articles were imported into referencing software EndNote (Programmer—The EndNote Team, Year—2013 Title—EndNote Place Published—Philadelphia, PA Publisher—Clarivate Version: EndNote 20 Type: 64 bit). Deduplicated and exported to Rayyan (Harvard, USA)[52]. Article screening by NL and JD with HA resolving any conflicts.

### Inclusion and exclusion criteria
Relevant primary articles addressing global patient co-ownership, electronic health records (EHRs), Personal Health Records (PHR) and includes data-co-ownership, patient rights, ethics, economics of PHR patient systems, personal care records and patient outcomes from threatened security were identified. This included randomised controlled clinical trials performed on PHRs. Articles were included if they discussed cloud-based personal health records that had patient and

healthcare provider co-ownership. Abstracts, reviews, conference proceedings, articles that do not reference PHR systems and with unclear outcomes were excluded. Specific exclusions included lack of reference to patient co-ownership with and without cloud-based infrastructures.

Initial recording of the number of articles found. Then a transparent selection process by reporting on decisions made at various stages of the systematic review. Numbers of articles are recorded at the different stages.

## Meta-analysis

A meta-analysis was performed for PHR domains investigating efficiency, security, and cost-based parameters. This was based on access policies, runtimes, encryption and decryption times, key generation times, distributed network related data transfer cost (gas cost) and other time-based activities. Log-transformation was applied when necessary. Also, a ratio of means standards effect size estimation on encryption calculated using the following formulae (Mean of intervention−Mean of control)/Mean of control. Analysis was performed using STATA (StataCorp 2013, Statistical Software, Release 13 College Station, TX StataCorp LP) for random effects modelling due to result heterogeneity. Significance was set at a $p < 0.05$. Authors contacted for completion of data if unclear or incomplete.

## Validity and bias

Risk of bias (ROB) with seven domains (D1-D7) of ROBINS-I and RobVis tools. Inclusion and exclusion criteria design to minimise bias. Search strategy disagreements resolution. Publication bias assessment with Egger's test and no adjustments necessary. PRISMA-guided protocol. PROSPERO registered review protocol for transparency and reduced bias. Manual check of all retrieved articles assessed the quality of included studies. Evaluation of the risk of bias in each study. Comprehensive search strategy. Wide time frame and scope of multiple databases, ensuring all relevant studies identified. Clearly defined inclusion and exclusion criteria established to select studies. Criteria applied consistently to reduce selection bias. Standardized data extraction forms and protocols to collect relevant information from each included study. To assess the risk of bias within individual studies, quality assessment tools mentioned above. Evaluation in all studies on PHR security checking on study design, data collection methods, and reporting quality. Assessment of heterogeneity controlled with a random-effects model Meta-Analysis methods with pooled effect estimates. Sensitivity with reanalysis. Robustness assurance. Reporting with PRISMA for transparent and complete reporting of methods and results, reducing reporting bias. The blinded screening and selection on Rayyan made by two reviewers and arbitration by a third. All unclear articles discussed, reaching consensus. For missing data, the respective article authors contacted for completion. No imputation necessary. Non-retrieved articles were 36. Their exact numeral contents unknown. Considered separately and deemed not meta-analysable. The results detailed in the RobVis tool in Fig. 5, visualising the risk-of-bias domains for each included study, see results. See further search strategy details in Supplementary S1, PRISMA checklist in S3.

## Reporting summary

Further information on research design is available in the Nature Portfolio Reporting Summary linked to this article.

# Data availability

The data generated in this study are provided in the Supplementary Information. Source data are provided with this paper. Source data and raw data generated in this study, have been deposited in the article repository on Figshare, https://doi.org/10.6084/m9.figshare.c.7066553. All data are available on the repository without restrictions. The timeframe for response to requests is immediate. All data are free to use.

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

## Acknowledgements

This study was supported by the Swedish Research Council (2019-01157) and the Swedish National Heart and Lung (20180505) and Freemasons Children's House foundations grants to Prof Eric Herlenius and scholarship to Dr Niklas Lidströmer. We acknowledge the librarians at Karolinska Institutet Narcisa Hannerz and Anja Vikingson, Professor Sabine Koch at Karolinska Institutet, the librarians at Imperial College London, Michael Gainsford, Sarah Feehan and Jackie Kemp.

## Author contributions

Niklas Lidströmer (NL) conceived the background research, idea and concept. NL and Joseph Davids (JD) designed the study. NL conducted the literature review with support from JD. NL performed data collection. NL and JD performed data analysis. NL assembled and structured the source data for the meta-analysis. All authors (NL, JD, Mohamed ElSharkawy (ME), Hutan Ashrafian (HA), Eric Herlenius (EH)) contributed to the data interpretation. HA and EH provided critical intellectual input throughout the study. All authors conducted statistical analyses and contributed to the interpretation of results. NL wrote the manuscript with input from all co-authors. NL made all revisions of the manuscript

with input from EH. All authors critically reviewed and approved the final version of the manuscript. NL created all figures and assembled all source data into a repository on Figshare.

## Funding

## Competing interests
The authors declare no competing interests.
