## [Peer Review File · Nature Communications]

Reviewers' Comments:

Reviewer #1:

Remarks to the Author:

The article systematically reviews sharing of electronic health records in a cloud. The text is original, follows PRISMA protocol, and presents exciting insights to the scientific community. The authors considered the most significant articles on the subject. The protocol is adequately done and gives a discussion pointing out the main challenges and issues.

Following, I list some suggestions to improve the article:

- GPOC is not a universally accepted term. It is a term that the authors have used in previous publications. Although I do not see a problem using this term in the article, the authors could better define what it means and why it could not use just PHR to represent this idea. Another doubt is what is a co-owned cloud. Is it a public cloud? Is it co-owned by the patient and its health providers?
- Pg. 3, line 45: The authors discuss centralized approaches. It would be interesting to mention fog-based and peer-to-peer/hierarchical methods for completeness.
- Pg. 3, line 63: How to deal with the language barrier among cross-border travel? I would appreciate a discussion on this subject.
- When discussing security and privacy, one of the focuses of the article, you could analyze the implications of HIPAA and GDPR. There are many aspects that these kinds of legislation affect privacy regarding health records and their relation to cross-border travel.
- Security is mainly discussed in terms of encryption and decryption of data. A more comprehensive analysis is needed, including the three main aspects of confidentiality, integrity, and availability.
- The method is well-defined and based on PRISMA. I did not find the range period for the search, the bases used, and the specific terms (on page 5, line 94, you only mention some keywords).
- Interoperability is an important issue that is not discussed at all. What is the rationale, and why was it not considered in the article? A global trend of adopting HL7 FHIR and other standards could influence the field.
- Security-based parameters are based on times in ms: time for encryption, time for decryption, and the ratio of means. These times depend on the infrastructure: CPU, memory, network bandwidth, etc. Just showing the number of effective time adds little value since it depends on the infrastructure used.
- I missed a big picture of the main issues in the discussion section. The discussion is based on the more traditional "Author X studied that; Author Y discusses." You could start with a big picture presenting the main topics and then more implicitly bringing the related studies.
- You cite types of encryptions on lines 281-282 (page 14). A discussion could be added because it could influence many essential aspects of the study.
- Blockchain is cited throughout the article, and I consider it a trend. You could discuss this technology further and its impact with more emphasis. Other crucial subjects still need to be explored, including Federated Learning, Fog Computing, and the Internet of Things (how to combine data from wearables from patient EHR, for instance).
- On page 19 you discuss machine learning and the use of data for decision, preserving privacy, and not disclosing patients' details. Anonymization and obfuscation, among other methods, could be commented on here. Although machine learning is not a focus of the article, it could influence GPOC in the services provided and organization/storage of data.

- Line 386, Pg. 20, you cite "four main areas," but it seems to be five. It would be nice to see some future directions.

Reviewer #2:

Remarks to the Author:

I appreciate and congratulate all authors for adding knowledge to the Global Patient co-Owned Cloud (GPOC) a Cloud-based infrastructure that shares information in the Personal Health Records (PHRs) space. This topic is timely and requires significant effort to establish particularly the concern on data security and privacy. I believe the meta-analysis and results presented cover the area to some extent however the term "use-ubiquity globally" that appears in the abstract does not take any advantages and/or support to the claims (refer to the conclusion).

Some areas to improve based on the Electronic Health Records (EHRs) use are:

(1) Validity and Bias:

This is very critical to explain how the risk minimisation was managed.

(2) Perhaps, the language tone could be lower to support the general audience and/or readers (since this is Open Access Forum);

(3) In addition, Blockchain protocol-embedded PHRs might be another approach that did not dominate the claim hence expanding such idea and thinking would be an added advantage to the paper.

(4) The conclusion section:

I believe there are standards when sharing EHRs. Such standards are depending on the Geographical locations, the Country's data protection, and/or breaches of legislation and their maturity (this is a very challenging area indeed).

ADDITIONAL COMMENTS:

The manuscripts share some form of information from the literature search. While they are interconnected, scientifically it is appropriate and valid to share the information however, the merit of the research contributions is somewhat diluted. Hence, it would be advisable to realign the findings specifically to the papers titled, for the readers' comfort and benefits.

In general, the body of work completed was commendable. At the same time, the written language may not be palatable to a wider audience and readers. It would be advisable to make it simple where possible since there are several paragraphs confounded with each other.

One of the classic examples in the paper titled "Systematic Review and Meta-Analysis for a Global Patient co-Owned Cloud (GPOC)" uses the term "use-ubiquity globally" in the abstract. This term is misleading based on the conclusion.

Please see the below answers and actions to all the 30 points of the major revision.

- (1) The manuscript ***GPOC SYS-META Clean*** does not contain comments, it is a revised clean manuscript.
- (2) The manuscript ***GPOC SYS-META Comms*** - you find the actions labelled with comments #1-30.
- (3) The manuscript ***GPOC SYS-META Highlighted*** - you find all the changes underlined and highlighted.
- (4) For convenience, the manuscript ***GPOC SYS-META Trace*** (Word-tracked) is fully traceable from the previous version that you reviewed.

Comments	#	Part	Point by Point Answers & Actions
Editor comments found in the separate cover letter.	1-8	All	See author cover letter.
REVIEWER COMMENTS			
Reviewer #1 (Remarks to the Author)			
The article systematically reviews sharing of electronic health records in a cloud. The text is original, follows PRISMA protocol, and presents exciting insights to the scientific community. The authors considered the most significant articles on the subject. The protocol is adequately done and gives a discussion pointing out the main challenges and issues.	9	All	We thank the reviewer for constructive and helpful comments that has ameliorated the manuscript.
Following, I list some suggestions to improve the article: GPOC is not a universally accepted term. It is a term that the authors have used in previous publications. Although I do not see a problem using this term in the article, the authors could better define what it means and why it could not use just PHR to represent this idea. Another doubt is what is a co-owned cloud. Is it a public cloud? Is it co-owned by the patient and its health providers?	10	Abst. & Intro.	It is important to define the unicity of the GPOC concept. This has now been implemented into the abstract and introduction. Hence, now there is a better definition of GPOC, what the co-ownership is, and why it is trisected for legal reasons. It has also been highlighted why a just PHR cannot be used – the unicity of the GPOC concept of cloud, ownership, AI integration, blockchain, foundation, globality, sharing, independence, interaction, legal foundation status and substrate for development of ML and its global dissemination has been elaborated.
- Pg. 3, line 45: The authors discuss centralized approaches. It would be interesting to mention fog-based and peer-to-peer/hierarchical methods for completeness.	11	Disc.	These have now been implemented as valuable new angles. Now the fog-based and peer-to-peer/hierarchical methods are mentioned.

- Pg. 3, line 63: How to deal with the language barrier among cross-border travel? I would appreciate a discussion on this subject.	12	Disc.	This is now included in the discussion among the challenges and importance of AI integration into GPOC. We exemplify how inbuilt AI in the GPOC can solve language barriers.
- When discussing security and privacy, one of the focuses of the article, you could analyze the implications of HIPAA and GDPR. There are many aspects that these kinds of legislation affect privacy regarding health records and their relation to cross-border travel.	13	Disc.	Thank you, we have now added a comment about and discuss this. After this comment we refer to the GPOC Ethics article ⁴ in the reference list. This article focuses entirely on ethics, policy, legal, regulation, etc. This is a self-contained article within the GPOC series. GDPR & HIPAA now mentioned (without infringing into the Ethics' article), but now highlighted. Indeed, we cross reference to the article that focuses in this area.
- Security is mainly discussed in terms of encryption and decryption of data. A more comprehensive analysis is needed, including the three main aspects of confidentiality, integrity, and availability.	14	Disc.	Now included and analysed articles of the systematic review elaborating these main aspects of confidentiality (19), integrity (11), and availability (12). Within parenthesis the number of articles bringing this up.
- The method is well-defined and based on PRISMA. I did not find the range period for the search, the bases used, and the specific terms (on page 5, line 94, you only mention some keywords).	15	Meth. & Supp.	The PRISMA form and other search documents have now all been included as supplements and referred to as supplementary files S1 and S3. In S1 the full search strategy is visible, with search strings etc. In S3 the PRISMA checklist is presented.
- Interoperability is an important issue that is not discussed at all. What is the rationale, and why was it not considered in the article? A global trend of adopting HL7 FHIR and other standards could influence the field.	16	Disc.	Now interoperability has been implemented into the discussion. Now also HL7 FHIR is mentioned. Also, as cross referencing here has been made to the technical EcoTech Article . ²
- Security-based parameters are based on times in ms: time for encryption, time for decryption, and the ratio of means. These times depend on the infrastructure: CPU, memory, network bandwidth, etc. Just showing the number of effective time adds little value since it depends on the infrastructure used.	17	Disc.	Now this point has been added to the big picture of the results, as a decisive factor to consider and weigh in. The infrastructure for GPOC is further elaborated in the technical EcoTech Article . ² It is cross referenced to this now too.
- I missed a big picture of the main issues in the discussion section. The discussion is based on the more traditional "Author X studied that; Author Y discusses." You could start with a big picture presenting the main topics and then more implicitly bringing the related studies.	18	Disc.	Now the big picture has been added in an initial overview, but also in the discussion. All the places with the traditionally mention "author X studied Y", etc, have been altered and the result is a much better flow. Now, with an accompanying big picture before and after, it is more readable. As advised the big picture is presented mainly first. Also, the traditional author mentioning has been partly altered where possible now, so that the sentences are more neutral as flow better for the reader.
- You cite types of encryptions on lines 281-282 (page 14). A	19	Disc.	A discussion has now been added on encryption. Also, referring further the technical EcoTech Article . ²

discussion could be added because it could influence many essential aspects of the study.			
- Blockchain is cited throughout the article, and I consider it a trend. You could discuss this technology further and its impact with more emphasis. Other crucial subjects still need to be explored, including Federated Learning, Fog Computing, and the Internet of Things (how to combine data from wearables from patient EHR, for instance).	20	Disc.	Now a discussion on blockchain has been added. Federated Learning, Fog Computing, and the Internet of Things are mentioned, but also a reference for further reading of the technical EcoTech Article . ²
- On page 19 you discuss machine learning and the use of data for decision, preserving privacy, and not disclosing patients' details. Anonymization and obfuscation, among other methods, could be commented on here. Although machine learning is not a focus of the article, it could influence GPOC in the services provided and organization/storage of data.	21	Disc.	Now anonymization and obfuscation and also fully homomorphic encryption is mentioned. There is now also a reference for further reading, where this is elaborated, in the technical EcoTech Article . ²
- Line 386, Pg. 20, you cite "four main areas," but it seems to be five. It would be nice to see some future directions.	22	Disc.	It has now been changed to eight (8) areas. Moreover, the initial keywords of each numbered paragraph have been italicised for increased clarity and readability. Also, the number of directions has been expanded and are now also synched with the six problems statements mentioned before and also referred to in the GPOC series. Again, without any overlap.
Reviewer #2 (Remarks to the Author)			
I appreciate and congratulate all authors for adding knowledge to the Global Patient co-Owned Cloud (GPOC) a Cloud-based infrastructure that shares information in the Personal Health Records (PHRs) space. This topic is timely and requires significant effort to establish particularly the concern on data security and privacy. I believe the meta-analysis and results presented cover the area to some extent however the term "use-ubiquity globally" that appears in the abstract does not take any advantages and/or support to the claims (refer to the conclusion).	23	Abst.	Many thanks. And yes, the term "use-ubiquity globally" that appears in the abstract, has been deleted and the whole abstract has been completely rewritten. It is now within the required limits of <150 words. Decreased from 244 words. The abstract now starts to explain GPOC, present the series to get a context for the reader, and then focuses on the article and what it shows.
Some areas to improve based on the Electronic Health Records (EHRs) use are: 1) Validity and Bias: This is very	24	Res.	Now under the headline "Validity and Bias" it has now been explained how the risk minimisation was managed.

critical to explain how the risk minimisation was managed.			
(2) Perhaps, the language tone could be lower to support the general audience and/or readers (since this is Open Access Forum)	25	All	As the editor also requested (and referred to reviewer 2) above in comment #1, very sentence of the whole manuscript has been put under revision and all paragraphs and nearly all sentences have been altered or polished. Hence, the language is now more suitable to a general audience and the sentences are shorter.
(3) In addition, Blockchain protocol-embedded PHRs might be another approach that did not dominate the claim hence expanding such idea and thinking would be an added advantage to the paper.	26	Disc.	Now the approach with Blockchain protocol-embedded PHRs have been mentioned. Reference added for further reading of the technical EcoTech Article . ²
(4) The conclusion section: I believe there are standards when sharing EHRs. Such standards are depending on the Geographical locations, the Country's data protection, and/or breaches of legislation and their maturity (this is a very challenging area indeed).	27	Disc. & Conc.	Now a hint about this is added. This is the full subject and ranger of the Ethics Article ⁴ . This article fully delves into geographical variations of ethics, legislation, regulations and policies. It is an additional combinatory literature review and interview series on this subject. It could not have been included into the present systematic review and meta-analysis and is self-contained and stands on its own legs. But a cross referencing has been added with the mentioned hint that his is important, and worth an independent article. See also the last point 8 of the discussion summary.
ADDITIONAL COMMENTS			
The manuscripts share some form of information from the literature search. While they are interconnected, scientifically it is appropriate and valid to share the information however, the merit of the research contributions is somewhat diluted. Hence, it would be advisable to realign the findings specifically to the papers titled, for the readers' comfort and benefits.	28	Abst. & Intro	Absolutely. The overlaps have now been minimised. Hopefully eradicated. For instance, it is requested in the revision to delve into a technical or regulatory area, and these have now been added, but only very brief and with a reference to the article which has this area as its main subject. It is hence important that the required technical additions to the systematic review do not infringe or dilute other manuscripts. Note: all mentioning of subjects belonging to another manuscript has been deleted. Similarly other subjects belonging to other self-contained parts of the series have been deleted and referred to only once for information to the reader (ref 1,2,3,4). Hence, a total realignment has been carried out.
In general, the body of work completed was commendable. At the same time, the written language may not be palatable to a wider audience and readers. It would be advisable to make it simple where possible since there are several paragraphs confounded with each other.	29	All	The whole text has now been rewritten. The language has been improved and the text flows better. It is adapted to the general audience. The sentences are less complex and shorter. Easier terminology has been used, whenever possible. When comparing the previous version and the present article manuscript all paragraphs and a majority of sentences have been rewritten to ease the flow for the general audience, which is very important. The paragraphing has been changed to eradicate any confusion. Note that now both the abstract (in two short sentences) and the introduction's first paragraph briefly mention the GPOC concept and the context of the GPOC series. Hence, the reader will immediately know what to expect from the articles regarding scope and contents. At the onset the reader will be aware that the five GPOC series entities will display distinct and non-overlapping foci. Hence, this sets the tone and contributes to the realignment of the articles. Several paragraphs hence became superfluous and have been deleted.

One of the classic examples in the paper titled "Systematic Review and Meta-Analysis for a Global Patient co-Owned Cloud (GPOC)" uses the term "use-ubiquity globally" in the abstract. This term is misleading based on the conclusion.	30	Abst.	This term has been deleted. The abstract has been completely rewritten. It has also been shortened from 244 to the required 150 words maximum. All intricate sentences with complex syntax and prosody have been altered. The message is clearer. Sentences shorter. Language simpler.
---	----	-------	---

Reviewers' Comments:

Reviewer #1:

Remarks to the Author:

The authors have conducted all the modifications asked for in the previous review. The article is now ready to be accepted.